# Dynamic Protein S-Acylation in Plants

**DOI:** 10.3390/ijms20030560

**Published:** 2019-01-29

**Authors:** Lihua Zheng, Peng Liu, Qianwen Liu, Tao Wang, Jiangli Dong

**Affiliations:** State Key Laboratory of Agrobiotechnology, College of Biological Sciences, China Agricultural University, Beijing 100193, China; caoyezlh@cau.edu.cn (L.Z.); liupeng_0605@163.com (P.L.); l_qianwen@126.com (Q.L.); wangt@cau.edu.cn (T.W.)

**Keywords:** lipid modification, S-acylation, protein S-acyl transferase, thioesterase

## Abstract

Lipid modification is an important post-translational modification. S-acylation is unique among lipid modifications, as it is reversible and has thus attracted much attention. We summarize some proteins that have been shown experimentally to be S-acylated in plants. Two of these S-acylated proteins have been matched to the S-acyl transferase. More importantly, the first protein thioesterase with de-S-acylation activity has been identified in plants. This review shows that S-acylation is important for a variety of different functions in plants and that there are many unexplored aspects of S-acylation in plants.

## 1. Introduction

Most eukaryotic proteins are post-translationally modified, and this modification has profound effects on protein function. An important class of post-translational modifications is lipid modification, which anchors proteins to membranes or specific lipid rafts by attaching fatty acid chains to specific amino acid residues [1,2]. The three most common lipid modifications are *N*-myristoylation, prenylation and S-acylation. *N*-myristoylation, the addition of 14 carbon myristoyl groups to N-terminal glycine residues via an amide bond [1]. Prenylation is the addition of 15 carbon farnesyl or a 20 carbon geranyl-geranyl group to the CaaX C-terminal cysteine of the target protein [3]. S-acylation is the addition of 16-carbon palmitate or an 18-carbon stearate moiety covalently linked to a specific cysteine in a target protein via a thioester bond [4]. S-acylation is unique because it is reversible. Acylation/de-S-acylation cycles can regulate subcellular targeting, which affects protein function, impacting a variety of events in the plant life cycle [3,4,5,6]. A proteomic study identified approximately 600 putatively S-acylation proteins expressed in Arabidopsis root-derived callus culture using a biotin switch isobaric tagging for relative and absolute quantification [7]. Some of the identified proteins are known to be S-acylated, such as GPA1, calcium dependant protein kinases (CDPKs) and calcineurin-B like proteins (CBLs). The novel proteins identified included mitogen-activated protein kinases (MAPKs), receptor-like kinases, integral membrane transporters, ATPases, and soluble *N*-ethylmaleimide-sensitive factor-activating protein receptors (SNAREs) [4,7]. These results suggest that S-acylation impacts a very broad range of proteins with varied functions.

## 2. S-Acylation and Heterotrimeric G Protein Subunits

Heterotrimeric G proteins are important signal transduction components that are conserved in all eukaryotes. Heterotrimeric G protein complexes consist of one Gα subunit, one Gβ subunit and one Gγ subunit [8]. Arabidopsis expresses one Gα (GPA1), one Gβ (AGB1) and three Gγ (AGG1, AGG2, AGG3) proteins. Adjobo-Hermans et al. report that the *Arabidopsis* Gα subunit, GPA1, has a myristoylation site at the G2 position and a S-acylation site at the C5 position. These two kinds of lipid modifications target GPA1 to the plasma membrane [9] (Figure 1). Zeng et al. show that the *Arabidopsis* Gγ subunit, AGG2, may be associated with the plasma membrane, depending on its prenylation and S-acylation states [10]. Dual lipidation of Gγ targets the Gβγ dimer to the plasma membrane, where it can form a heterotrimer with Gα [9] (Figure 1). In the inactive state, Gα is GDP-bound. Upon activation, Gα exchanges GDP for GTP and subsequently dissociates from the Gβγ dimer. Both subunits can activate various downstream effectors in the cytosol. By studies of mutants, it has been demonstrated that the function of G proteins affects almost every aspect of plant life, including cell division, pathogen defense responses, and hormone signaling [11,12,13,14,15,16]. 

## 3. S-Acylation and Small G Proteins

Some small G proteins are also known to be S-acylated. ROP (Rho-related GTPase from plants) GTPases are a plant-specific subfamily of Rho GTPases, and the Arabidopsis genome contains 11 ROP GTPases [17,18]. Based on their amino acid sequences, ROP1 to ROP8 are type-I ROPs, and ROP9 to ROP11 are type-II ROPs [19]. Inactive AtROP6 is only prenylated in the CaaL box motif and accumulates in Triton-soluble membranes [19,20] (Figure 1). Activated AtROP6 is subject to transient *S*-acylation on two highly conserved cysteine residues (C21 and C156) in the catalytic G-domain, and it accumulates in lipid rafts [20,21] (Figure 1). Gas chromatography-mass spectrometry analysis has been used to demonstrate that AtROP6 is acylated by modification with both palmitic and stearic acids and that the modification by stearic acid is more abundant [20,22]. Acylation-deficient *atrop6* mutants can bind and hydrolyze GTP but reduce the effects on polar cell growth, the endocytic uptake of the tracer dye FM4-64, and the distribution of reactive oxygen species [21]. Similar to ROP6, ROP2 has two corresponding S-acylation cysteine residues and ROP2 is localized to the plasma membrane [23] (Figure 1). AtROP10 is tethered to the membrane by the S-acylation of cysteine residues in the GC-CG box and its polybasic domain [19,24,25] (Figure 1). Plasma membrane localization is crucial for its function, especially in terms of responses to ABA in seed germination, root elongation, and stomatal closure as well as in the induction of the expression of the transcription factor MYB2 [26]. Ara6 is a novel, plant-unique type of Rab GTPase. Ara6 has a myristoylation site at Gly2 and a S-acylation site at Cys3, myristoylation is a prerequisite for S-acylation [27]. N-terminal fatty acylation, nucleotide binding and the C-terminal region determine the subcellular localization of Ara6 on early endosomes, and Ara6 functions as a regulator of membrane fusion in the endocytic pathway [27,28] (Figure 1). 

## 4. S-Acylation and Proteins Involved in Ca^2+^ Signaling

Several proteins involved in Ca^2+^ signaling have been reported to be S-acylated (Figure 1). Thirty-four genes encode CDPKs in Arabidopsis. The N-termini of most CDPKs are predicted to be myristoylated and are followed by cysteine residues, which may be S-acylated [29,30]. A plasma membrane association was observed for AtCPKs 7, 8, 9, 16, 21, and 28 [31]. AtCPK1 is associated with the peroxisome, implying that it may be involved in the regulation of lipid metabolism [31]. AtCPK2 is localized to the endoplasmic reticulum (ER) [32]. MsCPK3 in *Medicago sativa* and OsCPK2 in rice are localized to the plasma membrane [33,34]. LeCPK1 in tomato has also been shown to have a membrane distribution [35]. In Arabidopsis, 10 CBL proteins form a complex network with distinct CBL-interacting protein kinase (CIPK) involved in stress-responsive activities [36,37]. The CBL1 protein undergoes myristoylation at the G2 position and S-acylation at the C3 position [38]. Additionally, the CBL1 protein is targeted to the ER after myristoylation in the cytoplasm and is then trafficked to the plasma membrane by S-acylation in the ER. Lipid modification is not required for CBL1/CIPK1 complex formation but is crucial for targeting this dimer to the plasma membrane [38]. CBL1 function in salt tolerance is interrupted by either a G2A mutation or a C3S mutation, demonstrating the importance of both lipid modifications [38,39,40]. The tonoplast localization of CBL2 is regulated by the S-acylation of three cysteine residues in its N-terminus, and these palmitoylated Cys residues are conserved among several tonoplast-localized CBLs, such as CBL3 and CBL6 [41,42]. 

## 5. S-Acylation and Proteins Involved in Pathogenesis

S-acylation is also found on some proteins that are involved in the response to pathogens (Figure 1). Remorins are plant-specific proteins and play important roles in biotic interactions [43,44]. A recent study reveals that OsREM1.4 is localized to lipid rafts by S-acylation. In the early stage of *Rice stripe virus* (RSV) infection, the RSV-encoded movement protein, NSvc4, can interfere with remorin S-acylation, resulting in a large accumulation of remorins in the ER and subsequent degradation by autophagy [45]. RPM1-interacting protein (RIN4) is involved in the signal transduction of two plant cell defense systems during Arabidopsis resistance to *Pseudomonas syringae* [46]. RIN4 is tethered to the plasma membrane by S-acylation or/and prenylation [47]. The *P. syringae* type III effectors AvrRpm1 and AvrB are also associated with membranes by S-acylation, and they can phosphorylate RIN4, leading to the activation of the R protein **Resistance to *Pseudomonas syringae pv. maculicola 1*** (RPM1) [48]. RIN4 is also targeted by the type III effector AvrRpt2, which drives RIN4 disappearance, leading to the activation of the R protein, Resistance to *Pseudomonas syringae* (RPS2) [47]. The Leu-rich repeat transmembrane receptor kinase FLAGELLIN SENSITIVE2 (FLS2) is essential for flagellin recognition in *Arabidopsis thaliana* [49]. FLS2 requires S-acylation to elicit a full and efficient response to flagellin presence rather than localization to the plasma membrane [7]. Likely, as is the case for activated ROP6, S-acylation promotes the accumulation of FLS2 in lipid rafts, thus making signal transduction more efficient.

## 6. S-Acylation and Transcription Factors

Our own findings reveal a novel mechanism for the nuclear translocation of an S-acylated transcription factor in response to stress [50] (Figure 1). This transcription factor is involved in the dehydration stress response in *Medicago falcate*, we refer to it as *MfNACsa*. When we observed the subcellular localization of MfNACsa, we were surprised to find that it was not located in the nucleus, as are other transcription factors without transmembrane domains. We showed that MfNACsa has a S-acylation site at the C26 position, controlling its localization to the plasma membrane. Generally, transcription factors exercise their regulatory function in the nucleus. Therefore, we investigated the localization of MfNACsa under dehydration stress using green-fluorescent protein (GFP)fusion proteins. With the expression of MfNACsa-GFP driven by the native promoter, the nuclear-localized protein was abundant in PEG-treated hairy roots. The possibility exists that the nuclear translocation of MfNACsa is dependent on de-S-palmitoylation. In vitro, we observed a significant increase in nuclear-localized MfNACsa-GFP after 2 h of treatment with 0.05 M hydroxylamine (NH_2_OH), which can reveal protein de-S-palmitoylation through the cleavage of thioester linkages between cysteine residues and palmitate chains [51]. In vivo, we deduced that the nuclear translocation of MfNACsa relies on de-S-acylation by the thioesterase MtAPT1 [50]. We have also profiled the transcriptome to identify target genes regulated by MfNACsa under dehydration stress. Among the significantly differentially expressed transcripts, some upregulated genes were involved in fatty acid metabolism, for instance, nonspecific lipid-transfer proteins, fatty acid metabolism associated 3-ketoacyl-CoA synthase 6 and long chain acyl-CoA synthetase, which have overlapping functions in plant wax and cutin synthesis [52,53,54]. Our hypothesis is that MfNACsa is translocated to the nucleus to activate these genes to increase wax and cutin synthesis, which would reduce water loss or change membrane fluidity under drought stress. This is a very worthwhile aspect to explore in the future.

## 7. Protein S-Acyl Transferases and Thioesterases

Protein S-acylation reactions are catalysed by protein S-acyl transferases (PATs) and can be reversed by thioesterases [55,56,57]. In *Arabidopsis*, there are 24 PATs. However, only the substrates of PAT4 and PAT10 have been identified [23,41]. PAT4 may be responsible for ROP2 S-acylation in root hairs [23]. Zhou et al. have shown that PAT10 mediated CBL2/3/6 S-acylation occurs at the tonoplast [41]. Loss of *PAT10* function resulted in hypersensitivity to salt stress. PATs are distributed in diverse membrane compartments, and their localization determines the specific substrates that they modify [1]. Little is known about the determinants that control the specific localization of PATs. MtAPT1 is the first protein thioesterase with de-S-acylation activity that has been found in plants [50]. In plants several genes encode for thioesterases, e.g., in rice 30, in Arabidopsis 17 and in *Medicago truncatula* 42 thioesterase genes have been identified. Whether these thioesterases have de-S-acylation activity and substrate-specificity is worth studying in the future.

## 8. Outlooking

It is interesting that only one protein thioesterase has been identified in plants, while there are many S-acylated proteins. There are several possible explanations for this observation: (1) some proteins may dissociate from the membrane in other ways; (2) one thioesterase may catalyse the de-S-acylation of several substrates; and (3) other thioesterases may be present in plants but have not yet been discovered. In addition, the substrate specificity of PATs deserves attention. With further studies, revealing the dynamic mechanism of S-acylation and the biological functions impacted will help us better understand the significance of this lipid modification in plants.

## Figures and Tables

**Figure 1 ijms-20-00560-f001:**
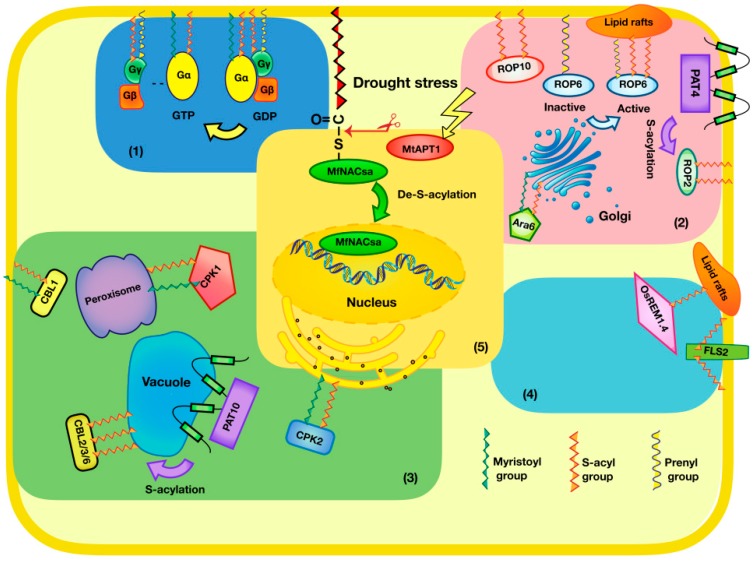
Some known S-acylated proteins in plants. S-acylation impacts a range of proteins with varied functions, including, heterotrimeric G protein subunits, small G proteins, proteins involved in Ca^2+^ signaling, pathogenesis related proteins and transcription factors. (1) Heterotrimeric G protein subunits. *Arabidopsis* Gα subunit, GPA1, is localized to the plasma membrane, depending on its myristoylation and S-acylation states. *Arabidopsis* Gγ subunit, AGG2, has a prenylation and a S-acylation. Dual lipidation of Gγ targets the Gβγ dimer to the plasma membrane, where it can form a heterotrimer with Gα. (2) Small G proteins. Inactive AtROP6 is only prenylated and accumulates in Triton-soluble membranes. Activated AtROP6 is subject to transient *S*-acylation and it accumulates in lipid rafts. ROP2 and ROP10 have two S-acylation cysteine residues. Protein S-acyl transferase 4(PAT4) may be responsible for ROP2 S-acylation in root hairs. A novel, plant-unique type of Rab GTPase, Ara6, has a myristoylation and a S-acylation. Ara6 is localized to early endosomes. (3) Proteins involved in Ca^2+^ signalling. AtCPK1 and AtCPK2 have a myristoylation and a S-acylation. AtCPK1 is associated with the peroxisome. AtCPK2 is localized to the ER. CBL1 associated with the plasma membrane, depending on its myristoylation and S-acylation. The tonoplast localization of CBL2/3/6 is regulated by the S-acylation. PAT10 mediated CBL2/3/6 S-acylation occurs at the tonoplast. (4) Proteins involved in pathogenesis. OsREM1.4 is localized to lipid rafts by S-acylation. FLS2 required S-acylation to elicit a full and efficient response to flagellin presence rather than localization to the plasma membrane. (5) Transcription factors. MfNACsa is a transcription factor and anchored to the plasma membrane by S-acylation. Under drought stress, MfNACsa is translocated to the nucleus after de-S-acylation by the thioesterase MtAPT1. This is the first report that a lipid-anchored transcription factor translocates to the nucleus by de-S-acylation in plants.

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
