# Peer review of "Dynamic Protein S-Acylation in Plants"

_ijms, 2019, doi:10.3390/ijms20030560_

Round 1

Reviewer 1 Report

I learned a lot by reading this short well written review.  I have no major criticisms. However, as someone who is not an expert in lipid PTMs I have some suggestions to make the manuscript more helpful to those in my situation:

Could a diagram be included, perhaps as part of figure 1, to show the actual chemical structure of the acyl cysteine thioester bond and the location of enzymatic cleavage by thioesterase.

Why is that multiple lipid modification tend to co-occur on the same protein? Consider commenting on this observation if any information is known.

Lines 20-22 “S-acylation is the only reversible lipid modification in which a 16-carbon palmitate or an 18-carbon stearate moiety is covalently linked to a specific cysteine in a target protein via a thioester bond [3]”  Is this not simply the definition of S-acylation?  It is not clear why you state it’s the only one.

Line 25-26: “global studies”  clarify what this means. Also explain why is S-acylation an important modification.

Line 33:  Give a brief definition of myristoylation (ie C14).

Line 46:  What is the nature of the transience of AtROP6 S-acylation?

Line 50-52:  I found this sentence awkward and difficult to understand.    Basal GTPase activity?

Line 52:  What is a potential S-acylation cysteine residue?  Is any cysteine a potential S-acylation site?

Line 73:  “trafficked”

Line 80:  How was is demonstrated that S-acylation occurred at the tonoplast.  Do you mean tonoplast localization of PAT10 is required to CBL 2/3/6 S-acylation? 

Line 102:  Does NACsa mean something, is it an acronym? 

Line 112-114:  citation needed after mtAPT1.

Line 127:  If absolute quantification was performed can you comment on the stoichiometry of these modifications (that the ratio of a given cysteine which is modified vs unmodified).  Stoichiometry is an often overlooked aspect of PTMs which can seperate PTMs with biological functions from random modifications.

Line 141: “proteins”

Line 142:  “factors”

Author Response

Dear  Reviewer, 

On behalf of my co-authors, Thanks a lot for your comments and suggestions concerning our manuscript entitled “Dynamic protein S-acylation in plants” (ijms-414986). These comments are all valuable and helpful for revising and improving our paper. We have studied comments carefully and have searched literature references to make a thorough revision. 

The responds to your comments are as follows.

Q1: “Could a diagram be included, perhaps as part of figure 1, to show the actual chemical structure of the acyl cysteine thioester bond and the location of enzymatic cleavage by thioesterase”.

R: We have shown the actual chemical structure of the acyl cysteine thioester bond and the location of enzymatic cleavage by thioesterase in our modified figure.

Q2: “Why is that multiple lipid modification tend to co-occur on the same protein? Consider commenting on this observation if any information is known”. 

R: It has been found that different lipid modifications have different strength to attach proteins to membranes. Modification with palmitoyl groups provides an affinity approximately 100 × stronger than a farnesyl group and 10 × stronger than a myristoyl group ( Hemsley., 2009, Molecular Membrane Biology ). Multiple lipid modification co-occur on the same protein, which may be more flexible in regulating the localization of proteins. 

Q3: “S-acylation is the only reversible lipid modification in which a 16-carbon palmitate or an 18-carbon stearate moiety is covalently linked to a specific cysteine in a target protein via a thioester bond [3]”  Is this not simply the definition of S-acylation?  It is not clear why you state it’s the only one”.

R: I'm sorry to find that our understanding of the definition of S-acylation is slightly biased after consulting the literature. And we have revised it in our manuscript.

Q4: ““global studies”  clarify what this means. Also explain why is S-acylation an important modification”.

R: “global studies” means that some proteomic studies have been carried out in eukaryotes (Hemsley et al., 2013, New Phytol, Kang et al., 2008, Nature, Roth et al., 2006, Cell ). According to reviewer 2, we changed the structure of the entire manuscript. Considering the logic of the manuscript, we have deleted this sentence in our manuscript.

Q5: “Give a brief definition of myristoylation”.

R: We have added the definition of myristoylation modification in the manuscript.

Q6: “What is the nature of the transience of AtROP6 S-acylation?”.

R: Sorek et al. show that upon GTP binding and activation, AtROP6, and possibly other ROPs, are transiently S-acylated, inducing their partitioning into lipid rafts. Lipid rafts may play an important role during polarity establishment in plants. The activation-dependent acylation of ROPs and their consequent partitioning in lipid rafts could play a role during the establishment of polarity in plants (Sorek et al., 2017, Mol. Cell. Biol ) .

Q7: “I found this sentence awkward and difficult to understand. Basal GTPase activity?”

R: I have modified this sentence in the manuscript. The meaning of this sentence is that acylation-deficient AtROP6 mutants can bind and hydrolyze GTP but reduce the effects on polar cell growth, the endocytic uptake of the tracer dye FM4-64, and the distribution of reactive oxygen species.

Q8: “What is a potential S-acylation cysteine residue?  Is any cysteine a potential S-acylation site?”.

R: ROP1 to ROP8 are type-I ROPs. Biochemical studies had previously shown that mutations of the corresponding residues in ROP6 abolished its S-acylation (Wan et al., 2017, Plant Signal. Behav ).

Q9: “How was is demonstrated that S-acylation occurred at the tonoplast.  Do you mean tonoplast localization of PAT10 is required to CBL 2/3/6 S-acylation?”.

R: Based on the results of the study, there are three evidences: (1) the tonoplast localization of CBL2 is regulated by the S-acylation, and mutating its potential palmitoylation sites rendered the protein cytosolic. The Cys residues for palmitoylation were conserved for CBL3 and CBL6; (2) localization of CBL2 and CBL6 at the tonoplast did not rely on vesicle trafficking (Batistic et al., 2012, Cell Res ), suggesting that their palmitoylation was conducted by tonoplast-localized PATs; (3) tonoplast localization of CBL2/3/6 was abolished by 2-BP treatment as well as in pat10 (Zhou et al., 2013, The Plant Cell ).

Q10: “Does NACsa mean something, is it an acronym”.

R: NAC is the acronym for NAM, ATAF1/2 and CUC2 and sa is the acronym for S-acylation.

Q11: “If absolute quantification was performed can you comment on the stoichiometry of these modifications (that the ratio of a given cysteine which is modified vs unmodified).  Stoichiometry is an often overlooked aspect of PTMs which can seperate PTMs with biological functions from random modifications”.

R: The proteomic study used a biotin switch isobaric tagging for relative and absolute quantification (iTRAQ)-based method to identify S-acylated proteins from Arabidopsis. Briefly, free cysteines were blocked with N-ethylmaleimide (NEM), and then S-acyl groups were removed by hydroxylamine treatment (Hyd+ sample) and exposed cysteines were tagged with biotin. Biotinylated proteins were purified on Neutravidin beads. As a control for nonspecific biotinylation and column binding, samples were split into two after NEM treatment and one-half was processed without hydroxylamine cleavage (Hyd- sample).Then, by using a 4-plex iTRAQ-based approach, researchers could compare protein abundance between experimental and control samples without the need to standardize to contaminant proteins, thereby reducing comparative errors. To increase confidence in the identification of S-acylated proteins, the entire sample preparation and mass spectrometric experiment was carried out twice, giving up to four biologically independent observations of the S-acylation state of a given protein ( Hemsley ., 2013, New Phytol ).

These are my responds to your comments.We would like to express our great appreciation to you  for comments on our paper.

Reviewer 2 Report

Lipid modifications are important post-translational modifications. In this short review, Zheng et al. pay attention on the S-acylation that is common in eukaryotes and anchors proteins to membranes or specific lipid rafts. In general, I think the review is worth to be published as it is the first of this kind in plants. However, I think this manuscript needs rewriting, changes in structure and reconfiguration of the figure. Therefore, I think it needs major revisions before being accepted for publication.

Major Comments:

1.       I think, the English used in this review can be improved; just some examples:

25/26: global studied have revealed S-acylation is…

27/28: have been experimentally demonstrated….

96: FLS2 requires???

34: Zheng et al showed

18: an important class of… is… lipid modification. Be more precisely in writing

And so on…

2.       I would suggest to improve the overall structure of the review and include further paragraphs and headings. First, I would describe the results of the proteomic story; and further I would separate the story into the target proteins regulated and the enzymes involved and identified so far

a.       Acylated poteins found in the proteomic approach

b.       Processes regulated by S-acylation

c.       Enzymes required and described

d.       Outlook (coming back to the proteomic approach: enzymes: why only one thioesterase for example) and the target proteins that could be analyzed.

3.       The overall structure of the figure is very unclear and confusing. On the one hand ROPs for example are presented between the heterotrimeric G proteins. On the other hand, ROP2 is presented between CBL1 and FLS2. This does not make sense and makes it very hard to follow the figure legends. I would suggest to separate the figure into five different boxes 1-5 to describe the several processes described in this review and regulated by S-acylation.

Minor comments:

-          12- Abstract: be more precisely- you do not mean different plant functions. Maybe functions in plants?

-          39/40 this study of mutants- which study- that has not been described earlier

-          41: new paragraph when beginning with small G proteins

-          I am not sure what the requirements of the journal are but I suggest mutants in short letters an italic: for instance, 50: atrop6

-          69: which kind of specific membrane?

Author Response

Dear  Reviewer, 

On behalf of my co-authors, we thank you very much for your positive and constructive comments and suggestions on our manuscript. We have tried our best to revise our manuscript according to the comments.  

The responses to your comments are as follows.

Q1: “I think, the English used in this review can be improved”.

R: We have revised our English grammar and awkward phrasing in hope of meeting the standard.

Q2: “I would suggest to improve the overall structure of the review and include further paragraphs and headings”.

R: We think that the structure of the whole manuscript you proposed is better than that of our original manuscript, so we have revised the structure of the whole manuscript and added subtitles according to your suggestion.

Q3: “The overall structure of the figure is very unclear and confusing”.

R: We have separated the figure into five different boxes to describe the proteins regulated by S-acylation in this review.

Q4: “Abstract: be more precisely you do not mean different plant functions. Maybe functions in plants?”

R: This is our mistake. We have corrected it in the manuscript.

Q5: “this study of mutants- which study- that has not been described earlier”.

R: This is our mistake and we have revised this sentence in the manuscript. 

Q6: “new paragraph when beginning with small G proteins”.

R: We have a new paragraph when beginning with small G proteins in the revised version.

Q7: “I am not sure what the requirements of the journal are but I suggest mutants in short letters an italic: for instance, 50: atrop6”.

R: This is our mistake and we have corrected it in the manuscript.

Q8: “69: which kind of specific membrane?”

R: MsCPK3 in Medicago sativa and OsCPK2 in rice are localized to plasma membrane (Gargantini et al., 2006, Plant J, Martin et al., 2000, Plant J ). LeCPK1 in tomato has also been shown to have a membrane distribution but it was not known what kind of specific membrane (Leclercq et al., 2005, J. Exp. Bot ). We have added details in the manuscript.

These are my  responses to your comments.

Round 2

Reviewer 2 Report

The revisions I asked for have been successfully addressed and I really appreciate the new, very clear structure of the manuscript.

However, I still think that at some point the language could be improved.

Some minor comments:

1.       Lines 70-72. Make one sentences out of two: … and ROP2 is localized to…

2.       Line 90: localized to THE plasma membrane

3.       Lines 158-160: In Medicago truncatula, there are 42 thioesterases.  Arabidopsis genome contains  17  thioesterases.  There  are  30  thioesterase  in  Rice.  Whether  these  thioesterases  have de-S-acylation activity and substrate-specificity is worth studying in the future. For the sentence in bold I suggest: In plants several genes encode for thioesterases, e.g in rice (should not be written with a capital letter) 30, in Arabidopsis 17 and in Medicago truncatula 42 thioesterase genes have been identified.

4.       Figure: in the legends the figure is divided into parts 1-5. This should also be shown in the figure!!!

Author Response

Dear  Reviewer,

On behalf of my co-authors, we thank you very much for your positive and constructive comments and suggestions on our manuscript. We have tried our best to revise our manuscript according to the comments.

The responses to your comments are as follows.

Q1: “Lines 70-72. Make one sentences out of two: … and ROP2 is localized to…”.

R: In our revised version, we have combined two sentences into one sentence.

Q2: “localized to THE plasma membrane”.

R: We have modified this sentence in the manuscript.

Q3: “Lines 158-160: In Medicago truncatula, there are 42 thioesterases.  Arabidopsis genome contains 17 thioesterases. There are 30 thioesterase in  Rice. Whether these thioesterases have de-S-acylation activity and substrate-specificity is worth studying in the future. For the sentence in bold I suggest: In plants several genes encode for thioesterases, e.g in rice (should not be written with a capital letter) 30, in Arabidopsis 17 and in Medicago truncatula 42 thioesterase genes have been identified.”. 

R: We think that the sentence you proposed is better than that of our original one, so we have revised the sentence according to your suggestion.  

Q4: “Figure: in the legends the figure is divided into parts 1-5. This should also be shown in the figure!!!”.

R:We have added the corresponding numbers in the figure to make it clearer.

These are my  responses to your comments.